# The Accuracy of Patient-Specific Spinal Drill Guides Is Non-Inferior to Computer-Assisted Surgery: The Results of a Split-Spine Randomized Controlled Trial

**DOI:** 10.3390/jpm12071084

**Published:** 2022-06-30

**Authors:** Peter A. J. Pijpker, Jos M. A. Kuijlen, Katalin Tamási, D. L. Marinus Oterdoom, Rob A. Vergeer, Gijs Rijtema, Maarten H. Coppes, Joep Kraeima, Rob J. M. Groen

**Affiliations:** 1Department of Neurosurgery, University Medical Center Groningen, University of Groningen, 9700 RB Groningen, The Netherlands; j.m.a.kuijlen@umcg.nl (J.M.A.K.); k.tamasi@umcg.nl (K.T.); d.l.m.oterdoom@umcg.nl (D.L.M.O.); ra.vergeer@umcg.nl (R.A.V.); g.rijtema@umcg.nl (G.R.); m.h.coppes@umcg.nl (M.H.C.); r.j.m.groen@umcg.nl (R.J.M.G.); 23D-Lab, University Medical Center Groningen, University of Groningen, 9700 RB Groningen, The Netherlands; j.kraeima@umcg.nl; 3Department of Epidemiology, University Medical Center Groningen, University of Groningen, 9700 RB Groningen, The Netherlands; 4Department of Oral and Maxillofacial Surgery, University Medical Center Groningen, University of Groningen, 9700 RB Groningen, The Netherlands

**Keywords:** spine surgery, virtual surgical planning (VSP), 3D-printing, patient-specific instrumentation, drill guides, computer-assisted surgery, image-guided surgery, image-guided navigation, pedicle screw, lateral mass screw

## Abstract

In recent years, patient-specific spinal drill guides (3DPGs) have gained widespread popularity. Several studies have shown that the accuracy of screw insertion with these guides is superior to that obtained using the freehand insertion technique, but there are no studies that make a comparison with computer-assisted surgery (CAS). The aim of this study was to determine whether the accuracy of insertion of spinal screws using 3DPGs is non-inferior to insertion via CAS. A randomized controlled split-spine study was performed in which 3DPG and CAS were randomly assigned to the left or right sides of the spines of patients undergoing fixation surgery. The 3D measured accuracy of screw insertion was the primary study outcome parameter. Sixty screws inserted in 10 patients who completed the study protocol were used for the non-inferiority analysis. The non-inferiority of 3DPG was demonstrated for entry-point accuracy, as the upper margin of the 95% CI (−1.01 mm–0.49 mm) for the difference between the means did not cross the predetermined non-inferiority margin of 1 mm (*p* < 0.05). We also demonstrated non-inferiority of 3D angular accuracy (*p* < 0.05), with a 95% CI for the true difference of −2.30°–1.35°, not crossing the predetermined non-inferiority margin of 3° (*p* < 0.05). The results of this randomized controlled trial (RCT) showed that 3DPGs provide a non-inferior alternative to CAS in terms of screw insertion accuracy and have considerable potential as a navigational technique in spinal fixation.

## 1. Introduction

Spinal instability is commonly treated through surgical fixation involving vertebral screw insertion. Conditions that frequently result in spinal instability are fractures after trauma, spinal deformities, tumors, and degenerative diseases. Spinal fixation is aimed at gaining stability and preventing subsequent neurological deficit. Generally, the bilateral positioning of screws in the vertebrae around the level(s) of instability is needed to achieve the immobilization of the unstable segments. The accurate insertion of screws is essential for achieving safe and optimal spinal fixation surgery. Conversely, malpositioned screws can induce damage of vital structures or result in the failure of fixation [1,2].

Traditionally, freehand insertion of screws is performed according to anatomical landmarks and through fluoroscopy control. However, because freehand screw insertion does not account for structural variations, including anatomical variance or severe deformations, there is an increased risk of malpositioned screws and related neurovascular complications [3,4,5,6]. Advances in spinal surgery have led to the development of computer-assisted surgery (CAS) navigation systems, also often mentioned as image-guided surgery, or intraoperative navigation. Initially, CAS systems relied on preoperative CT, which had to be manually re-registered for each individual vertebra and was associated with substantial registration errors. With the advent of modern CAS systems that rely on intraoperative acquired CT, this time-consuming repetitive calibration for individual vertebrae became redundant, leading to the increased use of CAS in spinal surgery. The accuracy of screw insertion has been substantially improved with current CAS systems, with a significantly reduced misplacement rate compared with the misplacement rate associated with freehand screw insertion [7,8]. Although spine surgeons with specific subspecialities (e.g., minimally invasive surgery and oncology) have widely incorporated CAS into their practices, the technology has not been adopted in all hospitals. In addition, the accuracy of screw insertion in cases with a highly mobile cervical spine reportedly fails to meet the high accuracy that is achieved in other spinal areas, which is most likely induced by the intraoperative shifting of segments. Moreover, despite these modern techniques, screw malpositioning does still occur, with the reported prevalence of malpositions being as high as 19% [9,10]. For these reasons, there is an ongoing demand for alternative navigational technologies that can be used to facilitate accurate spinal screw insertion.

Recent developments in medical computer-aided design and manufacturing techniques have given rise to completely new methods of surgical planning, commonly referred to as 3D virtual surgical planning (VSP) and patient-specific instrumentation (PSI) technology. The PSI technology comprises 3D-printed instrumentation that allows for the translation of the VSP to surgery. PSI are currently widely applied within different specialties [11], and the use of 3D-printed drill guides (3DPGs) has been attracting increasing attention as a promising navigational tool for spinal screw insertion. Recent studies have shown that 3DPGs are feasible for cervical and thoracic spine instrumentation, as demonstrated in cadaveric studies [12,13,14,15,16,17] and in clinical trials [18,19,20,21,22,23]. Moreover, their specific applications have been described for scoliosis surgery [24,25], C2 lamina screw insertion [26], and C1–C2 transarticular screw insertion [27]. Compared with CAS, the 3DPGs have several advantages. First, it provides the benefit of having a preformulated screw plan that includes the direction, length, and thickness of the screws. Second, it eliminates the need for intraoperative fluoroscopy. Third, it is applied to individual vertebra and is thus unaffected by intervertebral motion, which, in the case of CAS, can lead to workflow interruptions resulting from re-registration. Fourth, 3DPGs do not induce a surgical line of sight interference as in the case with CAS, which requires to constant looking back and forth between the patient and the screen. Last, 3DPGs do not require costly hospital investments.

The findings of various studies reported in the literature that have directly compared the accuracy of freehand and 3DPG screw insertion have all demonstrated the superiority of guides over the freehand technique [28,29,30]. However, comparisons of CAS and 3DPG are sparse within the literature; only one study has compared groups of patients instrumented with 3DPG and CAS [31]. To the best of our knowledge, a randomized comparative study of 3DPG and CAS has not been previously conducted.

Although 3DPGs appear to have several advantages, to become accepted as a viable alternative navigational tool its accuracy must first be shown to be comparable to that of CAS. Consequently, the objective of this randomized trial was to demonstrate the non-inferiority of 3DPG relative to CAS in terms of the accuracy of pedicle and lateral mass screw insertion.

## 2. Materials and Methods

### 2.1. Study Design

The SpineGuides study was a single-center, prospective, investigator-driven study that randomly allocated screws to either 3DPG or CAS-assisted instrumentation in patients undergoing clinically indicated fixation surgery. All consecutive patients scheduled for cervical and/or thoracic spinal fixation surgery were eligible for this study. Exclusion criteria were: (1) patients aged below 16 years, (2) scoliosis surgery, (3) previous surgical history entailing laminectomy or the application of osteosynthesis material at the target levels, (4) urgent cases, and (5) unilateral instrumentation. This trial was undertaken after obtaining approval from the ethical board of the local medical institution (ref no. M19.229543). Written informed consent was obtained from each patient prior to their enrollment in the study. The trial has been registered on euclinicaltrials.eu with a registration number of 2022-500880-11-00.

The study was designed to determine whether the accuracy of screw insertion in spinal fixation surgery performed with the 3DPG navigational technique is non-inferior relative to that of screw insertion performed using the CAS technique. Because spinal bone geometry, density, and microstructure can vary widely among subjects, a “split-spine” design was selected for the study. The two navigational techniques were randomly assigned to either the left or the right side of the spinal column. The split-spine design removed the influence of interindividual variability in the study arms and also ensured that vertebral levels and screw insertion techniques (mass lateral vs. pedicle) were evenly distributed.

The two techniques were randomly assigned to the right or left side of the spinal column by generating balanced permutations via computer randomization. The randomization was constrained by blocks such that an equal number of techniques per side are obtained. The inclusion was limited to bilateral screw insertion, which resulted in a consistent number of screws at each vertebral level within both study groups. The randomization scheme was created using the online tool at randomization.com (http://www.randomization.com, accessed on 5 January 2019) prior to commencing the study, and randomization codes were enclosed in sealed envelopes.

### 2.2. Virtual Surgical Planning

Virtual surgical planning (VSP) was carried out in accordance with a previously developed workflow, and 3DPGs were fabricated according to our previously published blueprint [17,23]. The brief description of the VSP and PSI steps are as follows. First, the preoperative CT was imported into medical image data segmentation software (Mimics, Materialise, Leuven, Belgium). Threshold-based segmentation was performed for each vertebral level to obtain 3D anatomical models, and optimal trajectories were manually defined. Then, 3DPGs were designed and manufactured in polyamide in accordance with ISO 13,485 standards and sterilized for intraoperative use by autoclave steam sterilization. Knowledge of the allocation was concealed from the 3D-specialist (PP) in charge of planning screw trajectories and designing the guides.

### 2.3. Surgical Technique

The standard surgical procedure was applied, starting cranially and continuing caudally placing screws sequentially at each level. Because randomization was concealed from the 3D specialist, the 3DPGs were designed with bilateral drill holes. Therefore, they would only fit if screws had not yet been inserted at the level of interest. Accordingly, for each level to be instrumented, the protocol stated to start with the 3DPG-assigned side followed by the contralateral CAS-assigned side. The 3DPGs were positioned after performing meticulous removal of soft tissue. Pilot holes (2 mm) were drilled at high-speed using appropriate drill stops (Figure 1). In the case of thick screws, the trajectories were enlarged/expanded using a straight pedicle probe. The study protocol ensured that the 3DPG burr hole checking was not performed using the CAS system, in order to keep the study arms separate. For the CAS study arm, the screw trajectories were created according to the standard CAS guiding procedure applied at our neurosurgical department, which consists of several steps. The steps applied in the CAS-assisted screw placement were as follows: (1) entry point identification using the CAS pointer and its marking with a ball-tipped burr, (2) definition of an optimal trajectory using the pointer, and (3) creation of a burr hole through alternated probing (or drilling for lateral mass) and pointer-based checking until the desired trajectory was achieved. For the purposes of this study, the ultimate CAS trajectory was saved intraoperatively by positioning the CAS pointer in the drill hole and storing the trajectory’s coordinates within the system. By opening the saved data using the CAS cranial module (instead of the regular spine module) postoperatively, we were able to retrieve the trajectory’s coordinates in the digital imaging and communications in medicine (DICOM) format (Figure 2). The cone-shaped pointer tip ensured concentric positioning within the drill hole. During the procedure, no navigated drill or screwdriver was used, as these tools were not part of our center’s standard procedure, nor it was available within the collection of instruments. The CAS setting comprised a mobile Arcadis Orbic 3D fluoroscopy C-arm (Siemens Medical Solutions, Erlangen, Germany) combined with a Brainlab optical navigation system (BrainLab Curve, BrainLAB, Munich, Germany).

### 2.4. Outcome Measures

The accuracy of the primary radiological screw insertion was the main parameter assessed in this study. Accuracy was measured by the amount of deviation from the plan using two continuous variables: (1) deviation from planned entry point (in mm); and (2) the 3D angular deviation (in degrees) from the planned trajectory. For the 3DPG arm, the deviation was measured following level-by-level registration of the postoperative CT with the preoperative CT, which included the planned trajectories. For the CAS study arm, the postoperative CT was registered with the intraoperative CT, which included the stored CAS trajectories (Figure 3). As a result of the registration per vertebra (single-level registration of each vertebra), the patient’s alignment (supine vs. prone) and spinal mobility did not affect the final analysis.

During the analysis, computerized registration and measurements were performed wherever possible. In technical terms, this procedure entailed (1) automated surface-based model registration using the Iterative Closest Point algorithm and (2) the use of automatically fitted analytical cylinders at the screw positions. These cylinders were automatically placed over the screw objects to prevent any assessment bias. Screw segmentation was done using standardized Hounsfield thresholds to eliminate bias by segmentation. The 3D deviation analysis has previously been shown to have very high inter-rater reliability, with an intraclass correlation coefficient of 0.99 [23].

### 2.5. Statistics

The objective of this study was to assess the non-inferiority of the 3DPG navigational technique relative to CAS. The calculation of the sample size was based on preselected margins of non-inferiority: 1 mm for entry-point accuracy and 3° for angular accuracy. In order to obtain representative margins and in the absence of representative published 3D deviation data, we conducted a small, pilot 3D simulation in which we measured the maximum amount of screw rotation until pedicle wall breach occurred. The upper limit of the rotation in which 99% of screws fitted within the pedicle was calculated as 3.29°. The allowable margins of error for screw placement reported in the literature are around 1 mm and 5°, respectively, for translation and rotation [32,33]. Although the metrics reported in the literature and those obtained in our pilot experiment were slightly different, they were in line with our predetermined margins. Hence, we concluded that they justified our selected margins of non-inferiority.

The sample size of each group was calculated according to the accuracy data derived from our pilot study along with additional pilot data gathered during CAS-assisted surgery. Because the current study focused solely on radiological accuracy outcomes obtained through 3D deviation analysis and because we assumed that the screws were independent, we calculated the screw-based sample size. Applying our assumptions in the calculation of sample size, we found that 36 screws demonstrated 90% power for determining non-inferiority at a significance level of 0.05. To compensate for unanticipated problems such as loss to follow up and equipment malformation, we included a dropout of 10% and therefore aimed to include a minimum of 40 screws (20 in each study arm). Considering the average number of cervical screws used per patient in our center, we aimed to include 10 patients in the study. The power calculation was performed using PASS software (NCSS, LLC, Kaysville, UT, USA).

All of the accuracy data were presented as descriptive statistics, expressed as median and interquartile range IQR values for non-normal distributed parameters. Non-inferiority was assessed by calculating the mean and 95% CI values for the difference between the 3DPG and CAS using a one-sample t-test and comparing the limits of the CI with predefined non-inferiority margins. The decision to reject the null hypothesis was made by determining whether the upper bound of the CI crossed the non-inferiority margin.

The final statistical analysis was performed using the SPSS Statistics program (SPSS Version 23.0 for Windows, IBM, N, Armonk, NY, USA).

## 3. Results

### 3.1. Patient Characteristics

Between June 2019 and December 2020, all of the consecutive patients referred for multi-level cervical and thoracic spine fixation were prospectively enrolled. A total of 10 patients were initially enrolled to meet the sample size calculations. A loss of CAS trajectory data due to storage failure, which exceeded the calculated dropout rate, resulted in the enrolment of three additional patients after approval by the institutional board had been renewed. Altogether, the number of patients suitable for the final analysis was 10. Because the ultimate number of screws inserted per patients turned out to be higher than expected, the final number of screws per study arm was 30.

The mean age of all patients was 56 years (range 16–82) and 5 of the 10 patients were women. The cohort presented with a spectrum of indications, including degenerative disease, osteoporotic fractures, rheumatoid arthritis, Klippel–Feil syndrome, and tumor.

### 3.2. Descriptive Statistics

The median entry point deviation was 1.8 mm (IQR: 1.0 mm–2.9 mm) in the cohort instrumented with 3DPG, and 1.8 mm (IQR: 1.0 mm–3.2 mm) in the cohort instrumented with CAS. The angular deviation was 5.7° (IQR: 2.9°–9.1°) in the cohort instrumented with 3DPG and 5.3° (IQR: 3.8°–8.1°) in the cohort instrumented with CAS (Figure 4).

### 3.3. Non-Inferiority Assessment

The 95% confidence interval (CI) for the difference in means between 3DPG and CAS (3DPG-CAS) was −1.01 mm to 0.49 mm. Therefore, the entry-point accuracy of 3DPG demonstrated non-inferiority relative to CAS, as the upper margin of the CI did not cross the predetermined non-inferiority margin of 1 mm (*p* < 0.05), which has been visualized in Figure 5. For angular accuracy, the 95% CI for the true difference between the means was −2.30° to 1.35°. Therefore, the angular accuracy of 3DPG was also found to be non-inferior relative to CAS, as the upper margin of the CI did not cross the predetermined non-inferiority margin of 3° (*p* < 0.05) (Figure 5).

## 4. Discussion

In this prospective randomized clinical trial (RCT), we compared the accuracy of spinal screw insertion using 3DPG and CAS. Our results showed that screw insertion accuracy achieved using 3DPG was similar and non-inferior to that obtained with CAS.

To the best of our knowledge, this RCT is the first to compare the accuracy of spinal screw insertion using 3DPG and CAS. Previous studies have compared 3DPG and freehand screw insertion, with their findings leading to a general consensus that the 3DPG technology significantly reduces the incidence of pedicle screw malpositioning [29,30]. Moreover, significant reductions in radiation exposure and in the time taken for screw implantation using this technology have been reported [28]. We only found one study by Fan et al. that compared groups of patients instrumented with CAS and 3DPG [31]. This prospective cohort study compared the use of robot-assisted pedicle screw insertion with 3DPG, CAS, and free-hand fluoroscopy-controlled screw insertion. The study demonstrated that the accuracy of “acceptable” screw placement in the 3DPG-guided group (95.52%) was slightly higher than that in the CAS group (90.60%), with no significant difference found between the two groups. These results suggest that both techniques yield similar degrees of accuracy; however, a systematic, randomized comparison was not performed in the above study. It is generally acknowledged that pedicle screw insertion has been substantially improved through the use of CAS technology compared with free-hand screw insertion. Hence, it was our belief that this accuracy standard should be the reference for novel navigational technologies such as 3DPG. Therefore, to become accepted as viable navigational tool and to optimize safety, an RCT should be conducted between 3DPG and CAS with the aim of determining, at minimum, non-inferior screw insertion accuracy, as was done in the current study.

The results of our randomized study indicated similar degrees of accuracy for both techniques. Compared with the accuracy of CAS, that of 3DPG was non-inferior for both of the assessed parameters. In fact, the upper limits of our 95% CI were 0.50 mm and 1.35°, which were well below the respective non-inferiority limits of 1 mm and 3°. There was no indication of 3DPG being superior to CAS, as the CI upper limits were above zero. However, we believe that a sufficiently powered study could lead to a finding of the superiority of 3DPG in specific subgroups. In particular, the use of CAS in the highly mobile upper cervical spine may be associated with increasing errors when operating further away from the reference array, with surgical manipulation inducing slight realignments of vertebral levels [34]. As our study design and methodology for measuring accuracy differed considerably from those of Fan et al. (they did not report on quantitative differences between planned and actual screw directions), a comparison of the results of the two studies presents challenges [31]. Nevertheless, it can be concluded that again 3DPG evidences a high degree of accuracy and that the finding of the current study validates with more confidence that the accuracy of screw insertion using 3DPG is similar and non-inferior to that of CAS.

### 4.1. Implications of the Study’s Findings

In light of the findings of this study, 3DPG can be considered to be an effective and accurate alternative navigational technology relative to CAS for cervical and thoracic spine fixation. It is important to point out that the results obtained using 3DPG cannot be pre-guaranteed when implementing the same technique and that our surgical teams underwent a learning curve, performing several cadaveric surgeries that served as training sessions. Furthermore, our comprehensive point-of-care 3D planning and printing facility has evolved over time, and we have acquired sufficient professional knowledge and competent staff with extensive training, enabling us to guarantee high-quality performance standards in full compliance with the EU medical device regulation (EU 2017/745). Centers that lack these facilities should be made aware of the high-quality standards that are required, or they should find suitable commercial partners for VSP and PSI design. However, given the technology’s novelty, its commercial availability is currently limited.

Although 3D technology has great potential, the technique is not suitable for all cases. This particularly applies for trauma cases that require immediate fixation surgery. Since the here described technology needs pre-planning, manufacturing and sterilization of 3D-printed instrumentation, the whole process does at minimum require 3–4 days. Additionally, for minimal invasive approaches the current 3D technology is not yet suitable. However, minimal invasive screw insertion gains increased popularity in order to minimize tissue trauma. Three-dimensional printed guides for minimal invasive approaches remains largely and unexplored area. There are however few examples of which the SpineBox system is the most well-known [35]. Further studies are needed in this area to compare these new approaches with CAS technology.

### 4.2. Study Limitations

At our center, the CAS system was used in combination with a 3D fluoroscopy C-arm capable of acquiring an intraoperative cone beam CT scan. Current state-of-the-art CAS systems are, however, often installed with the newer O-arms, which potentially provide enhanced image quality. Although both systems offer high levels of accuracy, there is some evidence that the use of O-arms with CAS improves the level of accuracy [36]. Therefore, our results are not generalizable for all CAS set-ups. Consequently, future studies that entail direct comparisons of 3DPG and O-arm-equipped CAS are warranted.

Within current 3D fluoroscopy CAS systems, the screw trajectory is defined ‘on the spot’, and not according to a predefined screw plan. Therefore, in the present study, we could not measure the accuracy of CAS accuracy with respect to a preoperative plan; instead, we measured its accuracy using the saved, intraoperative acquired trajectories and CT image data. It is possible that the surgeon considered the trajectory associated with the acquired hole to be sufficient but not optimal. If accuracy is defined according to the extent of deviation from the most optimal trajectory, the study could entail a slight overestimation of the actual accuracy of CAS. This again exposes the major advantage of 3DPG; the optimal trajectory can be selected and considered preoperatively instead of being defined during the time-constrained and intensive period of surgery.

A consideration of more clinical variables, such as infection rates, intraoperative blood loss, duration of the operation, and radiation dosage, was beyond the scope of the current study design. Additionally, for analyzing subgroups with different screw techniques (lateral mass or pedicle) the study was insufficiently powered. Consequently, these variables were not included in the comparison of the two techniques. Therefore, prospective RCTs with larger sample sizes are still required for a comprehensive assessment of these two techniques. It is likely that higher-powered clinical trials are necessary to validate our findings with a higher degree of confidence and to evaluate whether the inclusion of other clinical parameters, such as surgery time (total/per screw), support the use of one or the other technique.

Our analysis of both end points was performed on a per-protocol basis. During the study, there was a drop out of 3 patients due to data loss, which should be prevented in future studies by multiple copies or cloud storage. In addition, some of vertebral levels that we had planned to include in the fusion were not instrumented in cases entailing a sufficient amount of fixation. Because these screws were not inserted, they could not be evaluated, making an intention-to-treat analysis impossible to perform. Therefore, only planned screws that were actually in situ and visible as postoperative image data were included in the analysis. Furthermore, in our opinion, an intention-to-treat analysis was not appropriate for the current study design because randomization pertained to the level of patient side rather than to patients. An intension-to-treat analysis would, therefore, only be necessary when for example the assigned sided were revered, which is something that did not occur in this study.

Within-patient clustering was not considered in this study. In the future, variance and thus confidence intervals need to be inflated to account for the effect of within-patient clustering for two main reasons. Firstly, because screws within patients are more likely to resemble each other than screws across different patients (violating the independence assumption) and secondly because treatment is assigned on the level of patient side, not on the level of the screw. Therefore, to accurately reflect dependencies in the data, cluster-randomized design (whereby patients are the clusters) will be used to appropriately power future studies.

## 5. Conclusions

Although the benefits of 3DPG and its accuracy have been repeatedly demonstrated, this is the first randomized controlled study that compares 3DPG with CAS. The results of this RCT indicate that the accuracy of spinal screw insertion using 3DPG is similar and non-inferior to that obtained with CAS. Future higher powered comparative studies should focus on studying specific subgroups of vertebral levels that have the potential to demonstrate superiority.

## Figures and Tables

**Figure 1 jpm-12-01084-f001:**
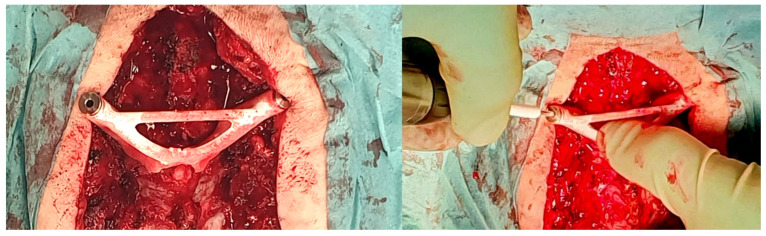
3DPG positioning and high-speed pilot hole drilling. A drill stop was slid over the drill bit to prevent penetration of the anterior cortex.

**Figure 2 jpm-12-01084-f002:**
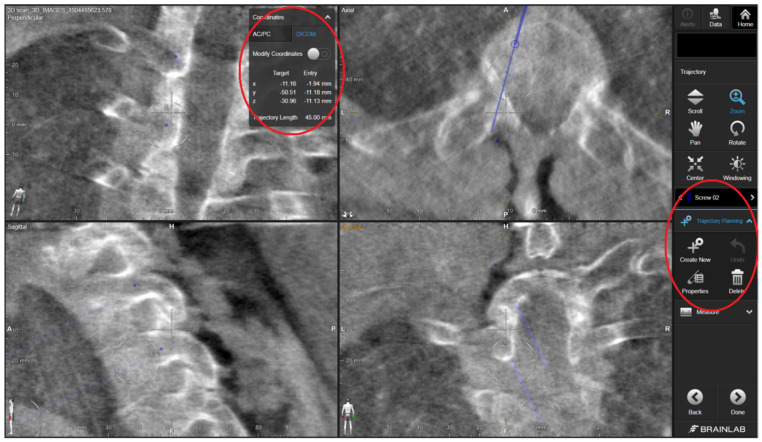
Retrieval of the DICOM coordinates of the intraoperatively stored trajectories accomplished by opening the spine planning file inside the cranial module of the planning software.

**Figure 3 jpm-12-01084-f003:**
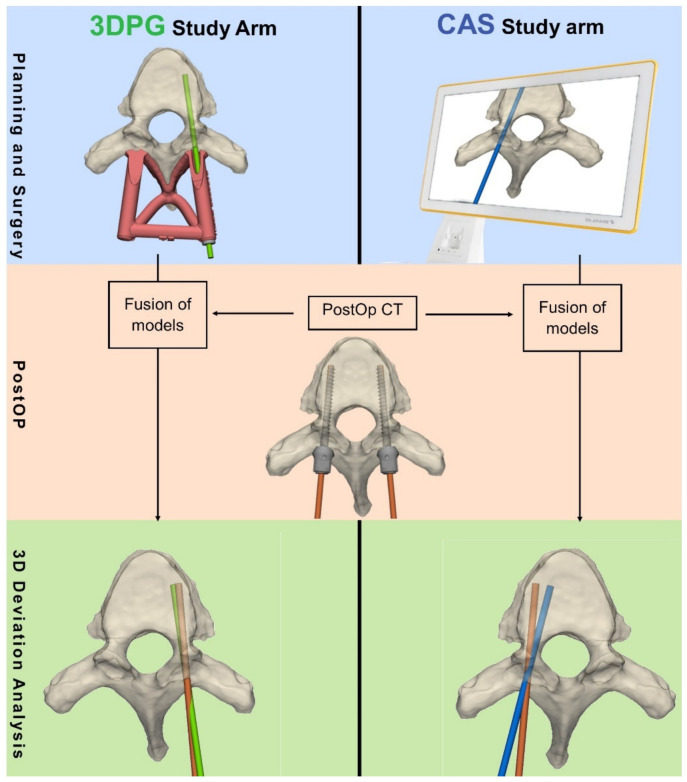
Schematic overview of the study depicting preoperative planning, preoperative measurements, and postoperative assessment procedures. 3DPG: 3D printed guides; CAS: Computer assisted surgery; 3D: 3 dimensional; PostOp CT: postoperative CT.

**Figure 4 jpm-12-01084-f004:**
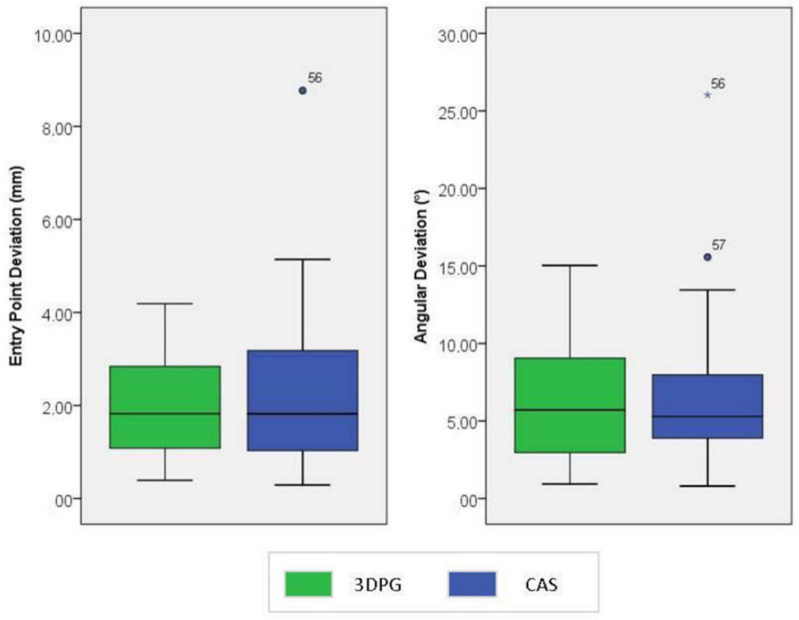
Entry point deviation (in mm) and angular deviation (in °) for screws used in the 3DPG (green) and CAS (blue) study groups.

**Figure 5 jpm-12-01084-f005:**
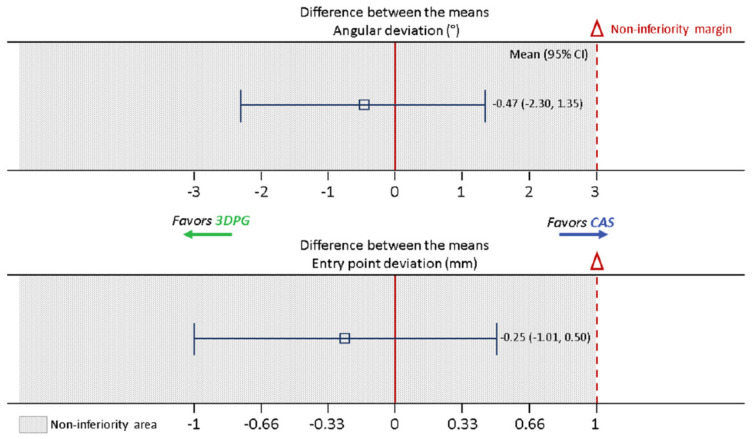
Graph displaying non-inferiority of 3DPG (test) relative to CAS (active control). The error bars demonstrate two-sided CIs, displaying both the lower and upper bounds of the CI. For both outcome measures, 3DPG was non-inferior relative to CAS, given that the entire CI was below the predetermined non-inferiority margins (Δ). It should be noted that a smaller outcome value (less deviation) indicated a better outcome. Therefore, areas to the left indicated better outcomes, and areas to the right indicated worse outcomes.

## Data Availability

The authors declare that the data supporting the findings of this study are available within the paper.

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
