# Peer review of "The Accuracy of Patient-Specific Spinal Drill Guides Is Non-Inferior to Computer-Assisted Surgery: The Results of a Split-Spine Randomized Controlled Trial"

_jpm, 2022, doi:10.3390/jpm12071084_

Round 1

Reviewer 1 Report

Thanks for your contribution to this interesting article; however, there are some concerns which should be modified:

1- In both methods and material and in exclusion criteria section, please explain comorbidity and chronic disease of patients

2- please put the "limitations of the study" at the end of discussion 

3- please rewrite the conclusion and delete limitations of the study from this section

Author Response

We wish to thank the reviewer for the review of our manuscript and for the generally positive feedback. The responses to your remarks are given below, with referrals to specific line numbers in  the track changes file where needed.

Comment 1:

In both methods and material and in exclusion criteria section, please explain comorbidity and chronic disease of patients

Answer:

We agree with the reviewer the description of the patient characteristics were somewhat limited. We have added detailed characteristics in line #239-241. Here we listed the patients’ age and the spectrum of indications for fixation surgery. Moreover we have rewritten the exclusion criteria more schematic, in lines #99-104

Comment 2:

please put the "limitations of the study" at the end of discussion 

Answer:

We have adopted the reviewer’s suggestion to move the study limitations towards the end of the discussion.

Comment 3:

please rewrite the conclusion and delete limitations of the study from this section

Answer:

We agree with the reviewer that the conclusion was written somewhat speculative. We have rewritten the conclusion accordingly.

Reviewer 2 Report

The authors describe a study comparing screw insertion occurred with CAS versus 3DPG. The paper is well written and is well introduced. Some concerns:

- the description of inclusion criteria needs to be more schematic and there needs to be a better description of how patients were selected;

- among the outcome variables it would deserve to compare blood loss and surgical times as well since the software instruction as is often the case with navigated screws sometimes takes a long time. If such data were not reported, however, the reasons should be described and included in a "study limitations" section.

- Has it been considered, at least for dorsal stabilizations, to compare with percutaneous insertion?

- The results section appears overly stringent; it would be helpful to add a summary table with the analysis and outcome differences reported;

- The conclusions are somewhat speculative and do not clarify the final outcome of the study;

Author Response

Our sincere thanks and appreciation for your extensive review of our paper and your positive general comments. We have modified the paper in response to your comments. The responses to your remarks are given below, with referrals to specific line numbers in  the track changes file where needed.

Comment 1:

the description of inclusion criteria needs to be more schematic and there needs to be a better description of how patients were selected;

Response:

Thank you for your suggestion. We agree and have rewritten the section that describes the inclusion criteria. Text has been added in #99-104. Patient were included consecutively based on these criteria.

Comment 2:

Among the outcome variables it would deserve to compare blood loss and surgical times as well since the software instruction as is often the case with navigated screws sometimes takes a long time. If such data were not reported, however, the reasons should be described and included in a "study limitations" section.

Response:

It has been a deliberate choice to omit the clinical outcome variables like blood loss and surgery time in this study for several reasons: 1) since this is still a novel technique it was deemed important to first find out how effective the technique is in achieving its primary goal: placing screws as accurate as possible with respect to the 3D plan. The accuracy is a directly related to the chosen technique. 2) what the subsequent effect on the clinical outcome is depends on a large number of factors. The screw insertion technique may only have minor influence. To be able to draw conclusions about these variables, the study size should be larger.  3) Since the study design did not allow for a double sided blinding, the surgery time variable could be (unintentionally) easily biased.

For all these factors it was chosen to keep this outside the scope of this study. These considerations have been described in the discussion in lines #399-#407

Comment 3:

Has it been considered, at least for dorsal stabilizations, to compare with percutaneous insertion?

Response:

It has not been considered in this study since the primary focus was cervical and upper-thoracic level posterior surgery. Minimal invasive screw placement is most often (at least in our center) intended for thoracolumbar or lumbal screw insertion in TLIF of PLIF procedures. The here described cases are most often accompanied by an laminectomy and therefore require an open procedure. Nevertheless; MIS screw placement mostly involves the use of CAS technology, and therefore the same screw insertion technology as was compared with in our study. Therefore we expect (although speculative) that a comparison between our open 3D guides and MIS CAS screw insertion would most likely result in the same conclusion (with regard to the screw placement accuracy). It would however be interesting to develop a 3D printed patient specific instrumentation for minimal invasive guide screw insertion and compare this to MIS pedicle screw placement assisted by CAS. We added some lines to the discussion about this potential for future research. #370-#379

Comment 4:

The results section appears overly stringent; it would be helpful to add a summary table with the analysis and outcome differences reported;

Response:

For our manuscript, we believe that it would be most helpful to the reader to support the results with figures rather than tables. We want to refer the reviewer to figure 4 and figure 5. We have consciously chosen to report our results by figures instead of tables since most reader are less familiar with non-inferiority study designs compared to superiority designs. A problem that we found during the revision is that the manuscript was the missing in-text reference to these figures. Adjustment to the text have been made in line #247, #252, and #256

Comment 5: 

The conclusions are somewhat speculative and do not clarify the final outcome of the study;

Response:

We agree with the reviewer, and have therefore rewritten the conclusion.

Round 2

Reviewer 2 Report

This is a well-conducted and easily studied and exploitable study for a good literature base.

I have some doubts about the title of the paper, which I suspect may be difficult to locate with a database search. My advice is to have good keywords and cite the free-hand techniques in the abstract.

Author Response

Thank you once again for reviewing our manuscript. We agree with the reviewer that it maybe difficult to locate the article when the search is specified on navigation technology. However, there simply isn't a uniform or generally accepted terminology for this. We have chosen for CAS, but others use 'image guided surgery,' or simply 'navigation'. Therefore it is indeed wise to broaden our keywords, we have adopted this suggestion. In addition we have also altered and added some text in #49 of the introduction.